# Pathological Response Predicts Survival after Pancreatectomy following Neoadjuvant FOLFIRINOX for Pancreatic Cancer

**DOI:** 10.3390/cancers15010294

**Published:** 2022-12-31

**Authors:** Hyun Jeong Jeon, Hye Jeong Jeong, Soo Yeun Lim, So Jeong Yoon, Hongbeom Kim, In Woong Han, Jin Seok Heo, Sang Hyun Shin

**Affiliations:** Division of Hepatobiliary-Pancreatic Surgery, Department of Surgery, Samsung Medical Center, Sungkyunkwan University School of Medicine, Seoul 06351, Republic of Korea

**Keywords:** pancreatic cancer, FOLFIRINOX, pancreatectomy, neoadjuvant therapy, neoplasm regression, adjuvant chemotherapy, neoplasm recurrence

## Abstract

**Simple Summary:**

In pancreatic cancer, complete pathologic response (cPR) after neoadjuvant treatment (NAT) has been rarely reported and its clinical course is not well known. Since the introduction of FOLIFIRINOX in clinical practice, the efficacy of chemotherapy has shown a dramatic improvement and these regimens have become the mainstay of chemotherapy. However, prior studies on cPR with various neoadjuvant regimens do not reflect the current trend. The aim of this study was to investigate the clinical course of patients according to pathological response, including cPR, who underwent resection following FOLIFIRNOX in advanced pancreatic cancer. We identified the value of pathological response as the prognostic factor for overall survival (OS) and disease-free survival (DFS).

**Abstract:**

Background: The clinical course of complete pathologic response (cPR) in pancreatic cancer after neoadjuvant chemotherapy is not well known. The aim of this study was to investigate the clinical course of patients according to pathological response, including cPR, who received only FOLIFIRNOX in advanced pancreatic cancer. Methods: Patients who underwent pancreatectomy after FOLFIRINOX for pancreatic ductal adenocarcinoma (PDAC) from 2017 to 2019 were retrospectively reviewed. cPR was defined as an absence of residual tumor on pathologic report. A nearly complete pathologic response (ncPR) was defined as a tumor confined to pancreas parenchyma, less than 1 cm without lymph-node metastasis. cPR and ncPR were assigned into a favorable pathologic response group (fPR). Kaplan–Meier method and Cox proportional hazard models were used for analysis. Results: Of a total 64 patients, 8 (12.5%) had a cPR and 8 (12.5%) had a ncPR. In the fPR group, median OS and DFS were superior to those of non-pathologic response group (more than 60 months vs. 38 months, *p* < 0.001; more than 42 months vs. 10 months, *p* < 0.001). On multivariable analyses, fPR and adjuvant therapy were independent prognostic factors for OS (HR: 0.12; 95% CI: 0.02–0.96, *p* = 0.05; HR: 0.26; 95% CI: 0.09–0.74, *p* = 0.01) and DFS (HR: 0.31; 95% CI: 0.12–0.86, *p* = 0.02; HR:0.31; 95% CI: 0.13–0.72, *p* = 0.01). Conclusions: pathologic response predicts survival after pancreatectomy following neoadjuvant FOLFIRINOX for pancreatic cancer, and adjuvant chemotherapy following neoadjuvant treatment might be beneficial for OS and DFS.

## 1. Introduction

Pancreatic cancer is a malignancy with a very poor prognosis. Surgical resection gives the best opportunity for a curative treatment. However, resectable cases account for only 20%–25% at the time of diagnosis [1]. Metastatic pancreatic cancer (mPC) and locally advanced pancreatic cancer (LAPC) have been mainly treated with chemotherapy. According to changes in the National Comprehensive Cancer Network (NCCN) guidelines from 2017, even patients with borderline resectable pancreatic cancer (BRPC) should receive chemotherapy as first-line treatment [2].

Gemcitabine monotherapy has long been considered as standard treatment for pancreatic cancer [3]. However, since the introduction of combination therapy with 5-fluorouracil, leucovorin, oxaliplatin, and irinotecan (FOLFIRINOX) in clinical practice, the efficacy of chemotherapy has shown a dramatic improvement. Recently, several studies have reported cases with conversion surgery from LAPC after primary chemotherapy [4,5,6]. Cases with curative resection and complete pathologic response (cPR) have been reported [7,8,9].

The effect of cPR on some solid cancers has been proven. cPR is considered a surrogate end point for survival in non-small-cell lung cancer [10,11]. In rectal cancer, cPR is a strong prognostic factor for long-term survival. Treatment guidelines are well established after acquiring cPR [12,13].

A cPR rate of 4%–15% in pancreatic ductal adenocarcinoma (PDAC) has been reported, which is lower than that in other solid tumors [14,15,16]. In addition, the clinical course of PDAC after acquiring cPR is not well known. Since the mainstay of chemotherapy has moved from gemcitabine monotherapy to FOLFRINOX and gemcitabine plus nanoparticle albumin-bound(nab) paclitaxel, few prior studies on cPR with various neoadjuvant regimens have reflected the current trend [17,18].

The aim of present study was to investigate the clinical course of patients according to pathological response, including cPR, who received only FOLIFIRNOX. We identified the value of pathological response as the prognostic factor for overall survival (OS) and disease-free survival (DFS).

## 2. Materials and Methods

### 2.1. Data Collection

Patients who underwent pancreatectomy after FOLFIRINOX for BRPC, LAPC, and mPC from January 2017 to December 2019 were analyzed retrospectively. The demographics of patients, including age, sex, presence of additional radiotherapy after chemotherapy, and cycles of chemotherapy were collected. Tumor characteristics included initial tumor size (mm), tumor location (head and body/tail), and TNM stage (American Joint Committee on Cancer 8). The baseline radiographic stage was classified into BRPC, LAPC, and mPC according to NCCN guidelines. The effects of chemotherapy for patients were evaluated every 4 cycles and recorded based on Response Evaluation Criteria In Solid Tumor (RECIST, version 1.1). Optimal Cancer Antigen 19-9 (CA19-9) response was defined if the value measured before operation was within the normal range regardless of the initial CA19-9.

Some patients received preoperative short-course radiotherapy after chemotherapy when the physician thought it would resolve the vessel involvement. In our institution, the patient treated with neoadjuvant chemotherapy underwent surgery within 5 weeks after the last date of chemotherapy. Additional radiotherapy was the only difference in management among the physicians, but the time until surgery after the last chemotherapy was the same.

Pathological response was classified into cPR, nearly complete pathologic response (ncPR), and non-pathologic response (non-PR) according to final pathology report. cPR was defined as an absence of residual tumor in the pancreas and lymph nodes on final pathologic report.

In general, statistically accessing the effect of cPR is difficult because of its small sample size. To overcome this hurdle, we defined ncPR by incorporating well-known indicator for long-term survival with the following criteria [19]: (1) tumor size less than 1 cm; (2) tumor was confined to pancreas parenchyma only; (3) no lymph-node metastasis; (4) R0 resection with more than 1 mm of free margin from malignant cells. Both ncPR and cPR were classified into a favorable pathologic response group (fPR).

OS was calculated from the date of the first chemotherapy administration to the date of death or the last date of follow-up. DFS was calculated from the date of surgery to the date of relapse confirmed radiologically. The recurrence pattern was classified into “loco-regional”, including remnant pancreas and/or near the surgical site, or “distant”, including para-aortic lymph node and/or peritoneal carcinomatosis and/or another organ metastasis.

### 2.2. Statical Analysis

All statistical analyses were performed using SPSS version 27.0 (IBM, Corp, Armonk, NY, USA). Baseline characteristics of the fPR group and the non-PR group were compared using the chi-square or Fisher exact test for categorical values and the independent *t*-test or Wilcoxon–Man–Whitney for continuous values. All categorical values are expressed as percentages. Continuous values are described as mean or median and interquartile ranges. Survival outcomes, including OS and DFS, between the two groups were compared using the Kaplan–Meier survival and log-rank test. The Cox proportional hazard model was used to find prognostic factors associated with OS and PFS. Factors with *p*-values less than 0.1 in univariate analysis were used for multivariate analysis. Statistical significance was considered when the *p*-value was less than 0.05.

## 3. Results

### 3.1. Baseline Characteristics

Sixty-four patients who underwent pancreatectomy after FOLFIRINOX for BRPC, LAPC, and mPC were identified. Their mean age was 61 ± 9 years. Males accounted for 53.1%. There were 45 (70.3%) cases of BRPC, 13 (20.3%) cases of LAPC, and 6 (9.4%) cases of mPC at diagnosis. A median of five cycles of chemotherapy were used before surgery. Radiotherapy between neoadjuvant chemotherapy and conversion surgery was performed for 17 (26.6%) patients. A fPR was observed in 16 patients (25%), including 8 cases of cPR (12.5%) and 8 cases of ncPR (12.5%).

A detailed comparison of baseline characteristics between fPR and non-PR patients is shown in Table 1. According to RECIST v1.1, complete and partial responses were observed in 24 (37.5%) patients, with the fPR group having more patients with complete and partial responses compared to the non-PR group (62.5% versus 29.2%; *p =* 0.02). Following chemotherapy, CA19-9 responses were observed in 33 (51.6%) patients, with the fPR group having significantly more patients with CA19-9 responses compared to the non-PR group (81.3% versus 41.7%; *p* = 0.01). The recurrence rate was significantly lower in the fPR group than in the non-PR group (33.3% versus 75%, *p* < 0.001).

### 3.2. Survival Outcome

The median follow up was 39 months (95% confidence interval (CI): 31–47 months). In the fPR group, the median OS could not be calculated because more than half of them were still living. The median OS of the fPR group was longer than that of the non-PR group (more than 60 months versus 38 months; *p* < 0.001; Figure 1A). The median OS of the cPR group and the ncPR group was superior to that of the non-PR, respectively (cPR versus non-PR (*p* = 0.04) and ncPR versus non-PR (*p* = 0.01) (Figure 1B)).

The median DFS of the fPR group was also superior to that of the non-PR group (more than 42 months versus 10 months, *p* < 0.001; Figure 2A). It was not significantly different in the cPR versus ncPR comparison (*p* = 0.17), nor in the ncPR versus non-PR comparison (*p* = 0.09) (Figure 2B).

### 3.3. Predictive Factor for Survival Outcome

The results of univariate and multivariate analyses of the predictors of OS are presented in Table 2. Negative resection margin (hazard ratio (HR): 4.65; 95% CI: 1.84–11.72; *p* < 0.001), adjuvant chemotherapy (HR: 0.26; 95% CI: 0.09–0.74; *p* = 0.01), and fPR (HR: 0.12; 95% CI: 0.02–0.96; *p* = 0.05) were independent predictors of OS.

Table 3 shows the results of univariate and multivariate analyses for factors associated with DFS. Adjuvant chemotherapy (HR: 0.31; 95% CI,0.13–0.72; *p* = 0.01) and fPR (HR: 0.31; 95% CI, 0.12–0.86; *p* = 0.02) were identified to be independent prognostic factors for DFS.

## 4. Discussion

This study revealed that a favorable pathologic response was associated with improved OS and DFS after neoadjuvant FOLFIRINOX. The median OS of longer than 60 months observed in our study was comparable with the 73.4 months to 76 months shown in MD Anderson Cancer Center and Johns Hopkins Hospital, which investigate the clinical course of cPR [16,20]. However, it was remarkable that even patients with ncPR presented a similar survival curve to those with cPR. A previous study used the same concept of ncPR as we did, and it reported that the median OS of ncPR was inferior to that of cPR (27 months versus more than 60 months, *p* = 0.003) [18]. The fact that we only enrolled patients treated with neoadjuvant FOLIFINOX might explain such different results. FOLFIRINOX and gemcitabine plus nanoparticle albumin-bound(nab) paclitaxel are common first-line therapies for advanced pancreatic cancer. However, there are still no clinical trials that determine the superiority between the two regimens. Some retrospective studies have reported better survival and higher resection rate with FOLFIRINOX [21,22]. Studies on cPR using various regimens have found that neoadjuvant FOLFIRINOX is an independent prognostic factor for long-term OS [17,18]. Thus, the prolonged survival pattern in the ncPR group shown in this paper might reflect the efficacy of FOLFIRINOX.

It is still controversial which factors are reflecting the prognosis after neoadjuvant treatment (NAT). The factors that could be considered as the prognostic marker are radiologic, biologic, and pathologic responses.

Some studies reported that post-CT after NAT overestimated unresectability [23,24]. CR Ferrone et al. reported post-FOLFIRINOX imaging suggested unresectability, but 92% of patients who underwent surgery had R0 resection. They concluded that after neoadjuvant FOLFIRINOX, imaging no longer predicts unresectability [25]. In NCCN guideline from 2022 version 1, the criteria for resection following NAT are suggested. Patients with BRPC or LAPC should be explored if their CA19-9 has decreased and radiologic findings do not demonstrate clear progression. However, even if radiologic response may not predict the resectability, it might be a prognostic factor. M lee et al. reported RECIST response and declined CA19-9 after NAT was associated with OS in patients with LAPC [26]. In this study, RECIST and CA19-9 response did not show statistical significance in multivariate analysis for prognostic factors for OS and DFS. Detailed characteristics of patients with cPR and ncPR in this study are presented in Table 4. The fPR group had more patients with complete response and partial response than the non-PR group did. All patients with cPR (100%) showed CA19-9 response, which dropped to the normal range following chemotherapy. Mark J Truty reported chemotherapy of more than six cycles, CA19-9 response, and major pathologic response highly predicted OS in patients treated with NAT for BRPC and LAPC. They presented the combination of individual factors had more predictive value than a single factor did [27]. The reason for the discrepancy with the results of prior studies may be associated with our small sample size. Since these factors are still clinically important as prognostic factors, large-scale studies are needed.

We identified that adjuvant chemotherapy was also an independent prognostic factor for OS and DFS. It is definite that adjuvant chemotherapy improves the survival after curative surgery in resectable pancreatic cancer. However, the role of adjuvant treatment following NAT and which subgroup most benefits from adjuvant chemotherapy is not well established. From several studies, subgroups that benefited from adjuvant chemotherapy after NAT are various, as follow: negative nodal status, low-grade histology, and negative margin status, and advanced T stages [28,29,30]. In this study, of the 56 patients who received adjuvant chemotherapy, only three patients (5.4%) could not complete treatment because of general weakness. Considering the low toxicity rate of adjuvant chemotherapy, it should be recommended. In the cPR group, only one case of recurrence was observed. This was surprising given that a substantial proportion of patients relapsed in a prior study. Lee et al. observed that 9 (10.4%) of 86 patients had a cPR, with 4 (44.4%) of these 9 cases showing systemic recurrence within one year [31]. Blair et al. reported that 30 (9.1%) of 331 patients had a cPR. In their study, the median DFS for patients with cPR was 29 months and recurrence were observed in 14 (48.3%) patients [20]. The low rate of recurrence in our institution might be due to a high proportion of adjuvant chemotherapy. All patients (100%) with cPR received adjuvant chemotherapy, which was noticeably higher than those (8.7%–48%) in previous studies [16,20,31]. Cloyd et al. searched cPR using the National Cancer Database and documented that only 49 (20%) of 244 patients received adjuvant therapy [17]. The possible importance of adjuvant treatment even in patients with cPR might be suggested by the relatively low rates of recurrence shown in this paper. Due to the small sample size, we could not perform a statistical analysis on the relationship between adjuvant chemotherapy and recurrence in patients with cPR. A large-scale study is needed to confirm this in the future.

This study has several limitations worthy of discussion. First, this was a retrospective study known to have possible selection bias. Documents for performance status, such as Karnofsky grade or European Cooperative Oncology Group scale, were not available. Considering that FOLFIRINOX was only recommended for patients with good performance status because of its toxicity, only physically healthy patients might have been included in this study. In addition, the number of patients with cPR was very small, and it included heterogeneous characteristics from BRPC to mPC. To investigate the role of pathological response as a prognostic factor, cPR and the concept of ncPR was used for statistical analysis. However, cPR is not equivalent to ncPR as they showed a different clinical course. This could be associated with the small sample size. We demonstrated that the adjuvant chemotherapy is a prognostic factor for OS and DFS in patients following NAT, but subgroup analysis was not feasible due to the small sample size. A large-scale multicenter study is needed. Lastly, the follow-up period was short. As the median follow-up period of this study was 38 months, the incidence of death or recurrence might increase in the future. Thus, appropriate follow-up studies are needed.

However, our study is still meaningful despite its small sample size in that we present the clinical course of patients with cPR as a prognostic factor for patients who received only FOLFIRINOX, compared to prior studies that included a heterogeneous neoadjuvant regimen. In addition, this study shows the latest clinical outcomes because we reflected patients who were treated in 2017–2019. Lastly, we identified that adjuvant chemotherapy was beneficial for the patients following NAT. The low recurrence rate in cPR patients after adjuvant chemotherapy will be discussed in the follow-up study.

## 5. Conclusions

In conclusion, pathologic response predicts survival after pancreatectomy following neoadjuvant FOLFIRINOX for pancreatic cancer, and adjuvant chemotherapy might be beneficial following neoadjuvant treatment.

## Figures and Tables

**Figure 1 cancers-15-00294-f001:**
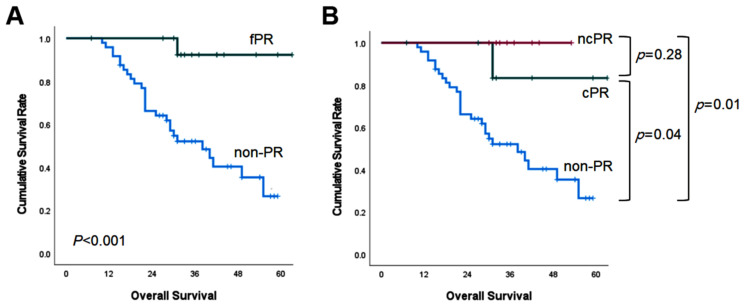
Kaplan–Meier survival curves of overall survival according to pathologic response.

**Figure 2 cancers-15-00294-f002:**
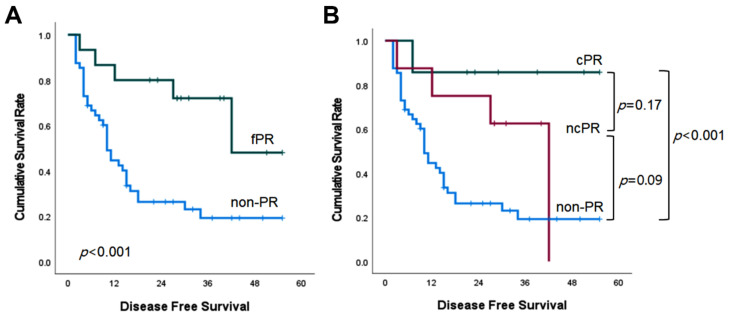
Kaplan–Meier survival curves of disease-free survival according to pathologic response.

**Table 1 cancers-15-00294-t001:** Clinicopathological characteristics (*n* = 64).

Characteristics	Overall Patients (*n* = 64)	fPR (*n* = 16)	Non-PR (*n* = 48)	*p*
Age	61 ± 9	63 ± 6	60 ± 10	0.35
Sex				0.15
Male	34 (53.1%)	6 (37.5%)	28 (58.3%)	
Female	30 (46.9%)	10 (62.5%)	20 (41.7%)	
Initial tumor size (mm)	32 ± 9	29 ± 10	33 ± 8	0.14
Disease stage				0.35
BRPC	45 (70.3%)	9 (56.3%)	36 (75%)	
LAPC	13 (20.3%)	5 (31.3%)	8 (16.7%)	
Distant metastasis	6 (9.4%)	2 (12.5%)	4 (8.3%)	
Cycle (median, range)	5 (4–8)	8 (5–11)	5 (4–8)	0.11
Additional radiotherapy (%)				0.74
No	47(73.4)	11(68.8%)	36(75.0%)	
Yes	17 (26.6%)	5 (31.3%)	12 (25%)	
RECIST response				**0.02**
CR/PR	24 (37.5%)	10 (62.5%)	14 (29.2%)	
SD/PD	40 (62.5%)	6 (37.5%)	s34 (70.8%)	
CA19-9 response				
Yes	33 (51.6%)	13 (81.3%)	20 (41.7%)	**0.01**
No	31 (48.4%)	3 (18.8%)	28 (58.3%)	
Vascular resection				0.55
No	40(62.5%)	11(68.8%)	29(60.4%)	
Yes	24(37.5%)	5(31.3%)	19(39.6%)	
Tumor location				0.53
Head	47(73.4%)	13(81.3%)	34(70.8%)	
Body/tail	17(26.6%)	3(18.8%)	14(29.2%)	
Lymph node status				**<0.001**
Negative	40(62.5%)	16(100%)	24(50%)	
Positive	24(37.5%)	0	24(50%)	
Margin status				0.19
Negative	56 (87.5%)	16 (100%)	40 (83.3%)	
Positive	8 (12.5%)	0	8 (16.7%)	
Adjuvant therapy				1
No	8(12.5%)	2(12.5%)	6(12.5%)	
Yes	56 (87.5%)	14 (87.5%)	42 (87.5%)	
Recurrence	41(64.1%)	5 (33.3%)	36(75.0%)	**<0.001**
Recurrence pattern				0.64
Loco-regional	17(41.5%)	4 (80%)	13(36.1%)	
Distant	24(58.5%)	1 (20%)	23(63.9%)	

Abbreviation: fPR = favorable pathologic response group; non-PR = non-pathologic response; BRPC = borderline resectable pancreatic cancer; LAPC = localized advanced pancreatic cancer; RECIST = response evaluation criteria in solid tumor; CR = complete response; PR = partial response; SD = stable disease; PD = progressive disease; CA19-9 = carbohydrate antigen 19-9.

**Table 2 cancers-15-00294-t002:** Analysis for the association between overall survival and clinical characteristics.

	Univariate Analysis		Multivariate Analysis	
Variables	HR (95% CI)	*p*	HR (95% CI)	*p*
Age		0.24		
<75	Reference			
≥75	0.30 (0.04–2.25)			
Sex		0.34		
Male	Reference			
Female	0.69 (0.33–1.47)			
Additional radiotherapy		0.97		
No	Reference			
Yes	1.02 (0.458–2.262)			
Cycle (median, range)		0.89		
<6 cycle	Reference			
≥6 cycle	1.05 (0.50–2.24)			
CA19-9 response		0.37		
No	Reference			
Yes	0.71 (0.33–1.50)			
RECIST response		0.20		
CR/PR	Reference			
SD/PD	1.72 (0.76–3.91)			
Resection margin		0.02		**<0.001**
Negative	Reference		Reference	
Positive	5.85 (2.36–14.51)		4.65 (1.84–11.72)	
Disease stage		0.21		
BRPC	Reference			
LAPC	1.17 (0.46–2.95)			
Distant metastasis	2.74 (0.90–8.35)			
Adjuvant therapy		0.04		**0.01**
No	Reference		Reference	
Yes	0.34 (0.13–0.93)		0.26 (0.09–0.74)	
Nodal status		0.01		0.25
Negative	Reference		Reference	
Positive	2.91 (1.37–6.17)		1.59 (0.73–3.48)	
Pathologic response		0.02		**0.05**
non-PR	Reference		Reference	
fPR	0.09 (0.01–0.63)		0.12 (0.02–0.96)	

Abbreviation: HR = hazard ratio; CI = confidence interval; CA19-9 = carbohydrate antigen 19-9; RECIST = response evaluation criteria in solid tumor; CR = complete response; PR = partial response; SD = stable disease; PD = progressive disease; BRPC = borderline resectable pancreatic cancer; LAPD = locally advanced pancreatic cancer; non-PR = non-pathologic response; fPR = favorable pathologic response group.

**Table 3 cancers-15-00294-t003:** Analysis for the association between disease-free survival and clinical characteristics.

	Univariate Analysis		Multivariate Analysis	
Variables	HR (95%CI)	*p*	HR (95% CI)	*p*
Age		0.55		
<75	Reference			
≥75	0.70 (0.21–2.28)			
Sex		0.15		
Male	Reference			
Female	0.63 (0.34–1.18)			
Additional radiotherapy		0.66		
No	Reference			
Yes	0.86 (0.43–1.72)			
Cycle (median, range)		0.88		
<6 cycle	Reference			
≥6 cycle	0.95 (0.51–1.78)			
CA19-9 response		0.06		0.68
No	Reference		Reference	
Yes	0.55 (0.29–1.02)		0.87 (0.45–1.70)	
RECIST response		0.66		
CR/PR	Reference			
SD/PD	1.16 (0.61–2.21)			
Resection margin		0.03		0.16
Negative	Reference		Reference	
Positive	2.47 (1.08–5.67)		1.82 (0.79–4.24)	
Disease stage		0.23		
BRPC	Reference			
LAPC	0.59 (0.23–1.51)			
Distant metastasis	1.875 (0.656–5.358)			
Adjuvant therapy		<0.001		**0.01**
No	Reference		Reference	
Yes	0.27 (0.13–0.69)		0.31 (0.13–0.72)	
Nodal status		0.12		
Negative	Reference			
Positive	1.64 (0.88–3.06)			
Pathologic response		0.01		**0.02**
non-PR	Reference		Reference	
fPR	0.27 (0.11–0.70)		0.31 (0.12–0.86)	

Abbreviation: HR = hazard ratio; CI = confidence interval; CA19-9 = carbohydrate antigen 19-9; RECIST = response evaluation criteria in solid tumor; CR = complete response; PR = partial response; SD = stable disease; PD = progressive disease; BRPC = borderline resectable pancreatic cancer; LAPC = locally advanced pancreatic cancer; non-PR = non-pathologic response; fPR = favorable pathologic response group.

**Table 4 cancers-15-00294-t004:** Details on the clinical course of patients with pathologic complete response and nearly pathological complete response after neoadjuvant treatment.

Case	Sex/ Age	Neoadjuvant Treatment (Cycle)	Additional Radiotherapy	Pretreatment Respectability	Initial Size (mm)	CA19-9 Response	RECIST Response	Operation	Pathologic Response	Adjuvant Treatment	Overall Survival	Recurrence Site	Disease-Free Survival	Status
1	M/60	12	(+)	BRPC	22	Yes	SD	Hepatopancreaticoduodenectomy	cPR	FOLFIRINOX	63 months	(−)	51 months	Alive
2	F/57	4	(−)	BRPC	23	Yes	PR	PPPD	cPR	Gemcitabine	32 months	(−)	29 months	Alive
3	F/56	10	(−)	BRPC	30	Yes	CR	PPPD	cPR	Gemcitabine	7 months			Follow up loss ^a^
4	M/63	4	(−)	LAPC	33	Yes	CR	PPPD	cPR	FOLFIRINOX	42 months	(−)	39 months	Alive
5	M/51	5	(−)	LAPC	22	Yes	SD	PPPD	cPR	Gemcitabine	27 months	(−)	23 months	Alive
6	F/74	11	(+)	LAPC	46	Yes	PR	PRPD	cPR	Gemcitabine	31 months	(+) peritoneum	7 months	Dead
7	F/64	14	(+)	Distant metastasis, para-aortic lymph node, Peritoneum	46	Yes	PR	Distal pancreatectomy	cPR	FOLFIRINOX	31 months	(−)	21 months	Alive
8	F/58	5	(−)	LAPC	30	Yes	PR	PRPD	cPR	FOLFIRINOX	59 months	(−)	55 months	Alive
9	F/63	5	(−)	BRPC	14	Yes	SD	Distal pancreatectomy	ncPR (8 mm)	Gemcitabine	44 months	(−)	40 months	Alive
10	F/60	5	(−)	BRPC	38	Yes	SD	PPPD/ SMV resection & anastomosis	ncPR (9 mm)	Gemcitabine	53 months	(+) Regional Lymph nodes	42 months	Alive
11	M/57	8	(+)	BRPC	18	No	SD	Distal pancreatectomy	ncPR (9 mm)	Gemcitabine	42 months	(+) solitary Lung	27 months	Alive
12	M/64	5	(−)	BRPC	22	No	SD	PPPD	ncPR (3 mm)	(−)	37 months	(+) solitary Liver segment 6	3 months	Alive
13	F/67	5	(−)	BRPC	30	Yes	PR	PPPD/ SMV resection & anastomosis	ncPR (6 mm)	Gemcitabine	35 months	(−)	31 months	Alive
14	F/70	6	(−)	BRPC	22	No	SD	PRPD/ SMV resection & anastomosis	ncPR (5 mm)	Gemcitabine	32 months	(−)	28 months	Alive
15	F/69	8	(−)	LAPC	31	Yes	PR	PPPD	ncPR (2 mm)	FOLFIRINOX	33 months	(−)	28 months	Alive
16	M/70	12	(+)	Distant metastasis, subclavicular, Para-aortic lymph node	43	Yes	CR	PRPD/ SMV resection & anastomosis	ncPR (5 mm)	(−)	30 months	(+) Lung, multiple	12 months	Alive

Abbreviation: CA19-9 = carbohydrate antigen 19-9; RECIST = response evaluation criteria in solid tumor; BRPC = borderline resectable pancreatic cancer; LAPC = locally advanced pancreatic cancer; CR = complete response; PR = partial response; SD = stable disease; PD = progressive disease; PPPD = pylorus-preserving pancreaticoduodenectomy; PRPD = pylorus-resecting pancreaticoduodenectomy; SMV = superior mesenteric vein; cPR = complete pathologic response; ncPR = nearly complete pathologic response. ^a^ the patient was identified as still alive, but she received all postoperative treatment at outside hospital. Overall survival was calculated as period of follow-up.

## Data Availability

The data presented in this study are available upon request from the corresponding author.

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
