# Peer review of "Pathological Response Predicts Survival after Pancreatectomy following Neoadjuvant FOLFIRINOX for Pancreatic Cancer"

_cancers, 2022, doi:10.3390/cancers15010294_

Round 1

Reviewer 1 Report

In this study, Hyun et al investigated the clinical course of patients undergoing a pancreatic resection after FOLFIRINOX and correlated pathological response to OS and DFS. In principle, this is an exciting and original study.

Major comments:

Since all patients after FOLFIRINOX were analyzed, I suggest changing the title of this manuscript.

Clinical Course of patients after pancreatectomy following Neoadjuvant FOLFIRINOX for Pancreatic Cancer 

OR

Pathological response predicts survival after pancreatectomy following Neoadjuvant FOLFIRINOX for Pancreatic Cancer

In order to make this study more homogeneous I would leave patients with metastatic disease out of this analysis. It is already very heterogeneous with some patients receiving radiotherapy and also adjuvant therapy differing considerably as did the type of surgery. 

Definition:

nCR was defined based on the following criteria:1) tumor size less than 1cm; 2) tumor was confined to pancreas parenchyma only; 3) no lymph-node metastasis, and 4) R0 resection with more than 1mm of free margin from malignant cells. Strictly spoken a tumor size of less than 1cm itself does not give any information regarding response to chemotherapy, neither N0 or R0. Could the authors comment on this? These could be favorable tumors on the forehand.

To make this manuscript easier to read, I would like to suggest the following abbreviations:

complete pathologic response (cPR) 

nearly complete pathologic response (ncPR) 

cPR together with ncPR is a favorable pathologic response (fPR)

non-pathologic response(non-PR)

By using these abbreviations, the authors should be consequent either to analyze 3 groups

cPR vs ncPR vs non-PR or fPR vs non-PR. I would strongly suggest combining cPR and ncPR because the number of patients is already quite low. So, all tables and KM -curves should be presented in this way. It is confusing to see both tables 1 and 2 with different columns. 

Then the conclusion should be:  After neoadjuvant FOLFIRINOX a favorable pathologic response (fPR)is a prognostic factor for OS and DFS after pancreatectomy for pancreatic cancer 

The conclusion: If appropriate adjuvant chemotherapy accompanied, recurrence rate 34 might be decreased is purely speculative and out of the scope of this article since it was not investigated with the proper number of patients

What could be of interest to the readers is the correlation between the PR and CA 19-9 dropdown or with the radiological response (which is known to be not accurate in these patients). Please provide this information.

Why some patients received additional radiotherapy? No information is given about when the radiotherapy was started. The same accounts for the time between the end of FOLFIRINOX and surgery nor for the time of surgery after radiotherapy.

Minor comments

FOLIFIRNOX is known to play a major role in advanced pancreatic cancer  This is already known by all of our readers so leave “known to play a major role in advanced pancreatic cancer” out every time.

The discussion is meant to discuss the current findings in this study and not to describe some patients in detail. Please discuss the number of cycles given of FOLFIRINOX or the role of radiotherapy in this category of patients. Make a suggestion on which adjuvant therapy should be given to the two groups

Author Response

Point 1: Since all patients after FOLFIRINOX were analyzed, I suggest changing the title of this manuscript.

Clinical Course of patients after pancreatectomy following Neoadjuvant FOLFIRINOX for Pancreatic Cancer 

OR

Pathological response predicts survival after pancreatectomy following Neoadjuvant FOLFIRINOX for Pancreatic Cancer

Response 1: Thank you for your thoughtful comments which helped in clarifying the aim of our study. I agree with you, and I changed my title

Pathological response predicts survival after pancreatectomy following Neoadjuvant FOLFIRINOX for Pancreatic Cancer

Point 2: In order to make this study more homogeneous I would leave patients with metastatic disease out of this analysis. It is already very heterogeneous with some patients receiving radiotherapy and adjuvant therapy differing considerably as did the type of surgery. 

Response 2: I deeply agree that our sample is heterogenous. but subject of our study includes very rare cases of pathologic complete response in pancreatic cancer. Pathologic complete response after neoadjuvant chemotherapy in pancreatic cancer was only studied as case reports in prior studies. Due to the nature of this study, heterogenous cases are unavoidable. Instead, I added limitation about this point that there were heterogeneous cases due to the small sample size.  

-> In addition, the number of patients with cPR was very small, and it included heterogeneous characteristics from BRPC to mPC. (Discussion, the 6th paragraph in page 12)

Point 3: nCR was defined based on the following criteria:1) tumor size less than 1cm; 2) tumor was confined to pancreas parenchyma only; 3) no lymph-node metastasis, and 4) R0 resection with more than 1mm of free margin from malignant cells. Strictly spoken a tumor size of less than 1cm itself does not give any information regarding response to chemotherapy, neither N0 or R0. Could the authors comment on this? These could be favorable tumors on the forehand.

Response 3:  As you pointed out, I agree that tumor size of less than 1cm cannot evaluate response to chemotherapy. However, this study enrolled patients with BRPC, LAPC or mPC at the time of diagnosis. We tried to present initial tumor size of all cohort and patients with ncPR in Table 1 and Table 5, respectively. Since their mean size was 32±9, tumor size of less than 1cm which was confirmed at final pathologic report can be considered as response to chemotherapy.  Because it is not likely to obtain N0 nor R0 in advanced cancer patients, acquiring N0 and R0 at final pathologic response was carefully interpreted that it had response to chemotherapy. In addition, considering definition of BRPC, LAPC included the vessel involvement, the cancer cell found only in pancreas parenchyma may reflect the shrinkage of tumor.

Point 4: To make this manuscript easier to read, I would like to suggest the following abbreviations:

complete pathologic response (cPR) 

nearly complete pathologic response (ncPR) 

cPR together with ncPR is a favorable pathologic response (fPR)

non-pathologic response(non-PR)

 By using these abbreviations, the authors should be consequent either to analyze 3 groups

cPR vs ncPR vs non-PR or fPR vs non-PR. I would strongly suggest combining cPR and ncPR because the number of patients is already quite low. So, all tables and KM -curves should be presented in this way. It is confusing to see both tables 1 and 2 with different columns. 

Response 4: I accepted all your thoughtful recommendation. Pathologic response was classified into fPR and non-PR. All abbreviations have been modified. This correction can be found throughout the paper.

Point 5: Then the conclusion should be:  After neoadjuvant FOLFIRINOX a favorable pathologic response (fPR)is a prognostic factor for OS and DFS after pancreatectomy for pancreatic cancer. The conclusion: If appropriate adjuvant chemotherapy accompanied, recurrence rate 34 might be decreased is purely speculative and out of the scope of this article since it was not investigated with the proper number of patients. What could be of interest to the readers is the correlation between the PR and CA 19-9 dropdown or with the radiological response (which is known to be not accurate in these patients). Please provide this information.

Response 5:  On multivariate analysis, fPR and adjuvant chemotherapy were prognostic factor. Thus, we concluded that pathologic response predicts survival after pancreatectomy following neoadjuvant FOLFIRINOX for pancreatic cancer. In addition, adjuvant chemotherapy might be beneficial for patients following NAT. In our opinion to correlation between the PR and CA19-9 dropdown was mentioned with reference.  

-> Some studies reported that post-CT after NAT overestimate unresectability. CR Ferrone et al. reported post-FOLFIRINOX imaging suggested unresectability, but 92% of patients who underwent surgery had R0 resection. they concluded after neoadjvuant FOLFIRINOX, imaging no longer predicts unresectabilitiy. In NCCN guideline from 2022 version 1, criteria for resection following NAT were suggested. Patients with BRPC or LAPC should be explored if their CA19-9 has decreased, and radiologic findings do not demonstrate clear progression. However, even if radiologic response may not predict the resectability, it might be a prognostic factor. M lee et al. reported RECIST response and declined CA19-9 after NAT was associated OS in patients with LAPC.

However, in this study, RECIST and CA19-9 response did not show statistical significance in multivariate analysis for prognostic factor for OS and DFS. Detailed characteristics of patients with cPR and ncPR in this study are presented in Table 4. fPR group had more patients with complete response and partial response than non-PR group. All cPR (100%) showed CA19-9 response which dropped to the normal range following chemotherapy. Mark J truty reported chemotherapy of more than 6cycles, CA19-9 response and major pathologic response highly predicted OS in patient treated with NAT for BRPC and LAPC. They presented the combination of individual factor had more predictive value than a single factor. The reason for the discrepancy with the result of prior studies may be associated with our small sample size. Since these factors are still clinically important as prognostic factors, large-scale studies are needed. (Discussion, the 4&5th paragraph in page 10-11)

Point 6: Why some patients received additional radiotherapy? No information is given about when the radiotherapy was started. The same accounts for the time between the end of FOLFIRINOX and surgery nor for the time of surgery after radiotherapy.

Response 6: All additional radiotherapy mentioned in this study are pre-operative short course radiotherapy. Additional radiotherapy was the only difference in management among the physicians, but the time until surgery after the last chemotherapy was the same.

-> Some patients received preoperative short course radiotherapy after chemotherapy when the physician thought it would resolve the vessel involvement. In our institution, the patient treated with neoadjuvant chemotherapy underwent surgery within 5 weeks after the last date of chemotherapy. Additional radiotherapy was the only difference in management among the physicians, but the time until surgery after the last chemotherapy was the same. (Materials and Methods, the 2nd paragraph in page 5)

Point 7: FOLIFIRNOX is known to play a major role in advanced pancreatic cancer.  This is already known by all of our readers so leave “known to play a major role in advanced pancreatic cancer” out every time.

Response 7:  I agree with you, and we deleted that phrase.

Point 8: The discussion is meant to discuss the current findings in this study and not to describe some patients in detail. Please discuss the number of cycles given of FOLFIRINOX or the role of radiotherapy in this category of patients. Make suggestion on which adjuvant therapy should be given to the two groups

Response 8:  Role for adjuvant treatment following neoadjuvant chemotherapy and which subgroup most benefited with adjuvant chemotherapy is unclear. However, this study presented that adjuvant chemotherapy after NAT and surgery was independent prognostic factor for improved OS and DFS. 3(5.4%) of 56 patients who received adjuvant chemotherapy after NAT couldn’t finish completion of adjuvant treatment due to poor condition. Considering the low toxicity rate of adjuvant chemotherapy, it should be recommended.

-> We identified that adjuvant chemotherapy was also an independent prognostic factor for OS and DFS. It is definite that adjuvant chemotherapy improves the survival after curative surgery in resectable pancreatic cancer. However, role for adjuvant treatment following NAT and which subgroup most benefited with adjuvant chemotherapy is not well-established. From several studies, subgroups which benefited from adjuvant chemotherapy after NAT are various as follow: negative nodal status, low-grade histology, and negative margin status, and advanced T stages. In this study, of the 56 patients who received adjuvant chemotherapy, only 3 patients (5.4%) could not complete treatment because of general weakness. Considering the low toxicity rate of adjuvant chemotherapy, it should be recommended. In the cPR group, only one case of recurrence was observed. This was surprising given that a substantial proportion of patients relapse in a prior study. Lee et al. observed that 9 (10.4%) of 86 patients had a cPR with, 4 (44.4%) of these 9 cases showing systemic recurrence within one year. Blair et al. reported that 30 (9.1%) of 331 patients had a cPR. In their study, the median DFS for patients with cPR was 29months and recurrence was observed in 14 (48.3%) patients. Possible reason for the low rate of recurrence in our institution might be due to a high proportion of adjuvant chemotherapy. All patients (100%) with cPR received adjuvant chemotherapy, which was noticeably higher than those (8.7%-48%) in previous studies. Cloyd et al. searched cPR using the National Cancer Database and documented that only 49 (20%) of 244 patients received adjuvant therapy. The possible importance of adjuvant treatment even in patients with cPR might be suggested by the relatively low rates of recurrence shown in this paper. Due to the small sample size, we could not perform a statistical analysis on the relationship between adjuvant chemotherapy and recurrence in patients with cPR. A large-scale study is needed to confirm this in the future. (Discussion, the 5th paragraph in page 11-12)

Reviewer 2 Report

Several limitations: small sample size, having associated nCR and pCR while the prognosis is very different (50% recurrence if nCR versus 13% if pCR).

Definite clinical benefit, in particular the probable benefit of performing adjuvant chemotherapy even if pCR. Data to be confirmed on a large prospective cohort, also including  resectable tumors.

Author Response

Point 1: Several limitations: small sample size, having associated nCR and pCR while the prognosis is very different (50% recurrence if nCR versus 13% if pCR).

Response 1: As you pointed out, the description of the difference in prognosis between nCR and pCR due to the small sample size was added to the limitation

-> In addition, the number of patients with cPR was very small, and it included heterogeneous characteristics from BRPC to mPC. To investigate role of pathological response as prognostic factor, cPR and the concept of ncPR was used for statistical analysis. However, cPR is not equivalent to ncPR as they showed different clinical course. This could be associated with small sample size. (Discussion, the 6th paragraph in page 12)

Point 2: Definite clinical benefit, in particular the probable benefit of performing adjuvant chemotherapy even if pCR. Data to be confirmed on a large prospective cohort, also including resectable tumors.

Response 2: We acknowledge the limitations of this study. The low recurrence rate in cPR patients after adjuvant chemotherapy will be discussed in the follow-up study.
